

# Efficacy of using Radar Induced Factors in Landslide Susceptibility Analysis: case study of Koslanda, Sri Lanka.

Ahangama Kankanamge Rasika Nishamanie Ranasinghe[1], Ranmalee Bandara[1], Udeni Gnanapriya Anuruddha Puswewala[2], Thilantha Lakmal Dammalage[1]

[1]Department of Surveying and Geodesy, University of Sabaragamuwa, Belihuloya, 70140, Sri Lanka
[2]Department of Civil Engineering, University of Moratuwa, Moratuwa, 10400, Sri Lanka

*Correspondence to*: A.K.R.N. Ranasinghe (nishamanie@geo.sab.ac.lk)

**Abstract.** Through recent technological developments of radar and optical remote sensing in the areas of temporal, spectral, spatial, and global coverage, the availability of such images either at a low cost or free of charge, and the advancement of tools developed in image analysis techniques and GIS for spatial data analysis, a large variety of applications using remote sensing and GIS as tools are possible. Hence, this study aims to assess the efficacy of using Radar Induced Factors (RIF) in identifying landslide susceptibility using bivariate Information Value method (InfoVal method) and multivariate Multi Criteria Decision Analysis based on the Analytic Hierarchy Process statistical analysis. Using identified landslide causative factors, four landslide prediction models as bivariate without and with RIF, multivariate without and with RIF are generated. Twelve factors topographical, hydrological, geological, land cover and soil plus three RIF are considered. The prediction levels of susceptibility regions are distinguished and categorized into four classes as very low, low, moderate, and high susceptibility to landslides. With integration of RIF, boundary detection between high and very low areas increased by 7%, and 4% respectively, and there is an improvement of 2.45% prediction and 1.12% validation performances of bivariate analysis than multivariate.

## 1    Introduction

Landslides are one of the major types of geo-hazards in the world as approximately 09% of global natural disasters are recorded as landslides (Chalkias et al. 2014). The recent statistics on landslide disasters per continent, from year 2000 to 2017, summarized in the Emergency Disaster Database (EM-DAT) indicate that landslides cause around 16500 deaths and affect 4.5 million people worldwide, with property damages of about US $3.5 million (OFDA/CRED 2016). The spatial prediction of landslide disasters, incorporating statistical analysis to identify areas that are susceptible to future land sliding, based on the knowledge of past landslide events, topographical parameters, geological attributes, and other possible environmental factors, is one the important areas of geo-scientific research (Park et al. 2013).



Presently, remote sensing technology has been used extensively to provide landslide-specific information for emergency managers and policy makers in terms of disaster management activities in the world (Baroň et al. 2014, Martha 2011). In recent years, there is an increasing demand for high resolution satellite data to be used for extracting geometric object information and mapping. The spatial resolution of space-borne optical data is now less than 1m in panchromatic images, and at the same time, the interest in Synthetic Aperture Radar (SAR) sensors and related processing techniques has also increased. Radar is considered to be unique among the remote sensing systems, as it is all-weather, independent of the time of day, and is able to penetrate into the objects. Additionally, radar images have been shown to depend on several natural surface parameters such as the dielectric constant and surface roughness. The dielectric constant is highly dependent on soil moisture due to the large difference in dielectric constant between dry soil and water (Kseneman et al. 2012). The forest and the vegetation cover of the earth surface is well sensed by the remote sensing techniques, where the shorter wave length regions as X and C radar bands identify the forest canopy clearly in radar remote sensing.

It is accepted in the scientific community that remote sensing techniques do offer an additional tool for extracting information on the causes of landslides and their occurrences. Especially for deriving various parameters related to the landslide predisposing and triggering factors at global and regional scales, remote sensing plays a vital role (Corominas et al. 2014, Muthu et al. 2008). Most importantly, landslide susceptibility analysis has greatly aided the prediction of future landslide occurrences, which is important for humans who reside in areas surrounded by unstable slopes. It is therefore identified that remote sensing techniques are significant in order to extract the landslide susceptibility regions by providing most suitable landslide predisposing factors at smaller scale.

It can be observed that there is massive potential for applicational research in the area of disaster management, if, conventional remote sensing data and radar are integrated. This is because each method has its inherent disadvantages and shortcomings, as well as advantages, and integrating the two could potentially complement each other. As such, this study combines the predisposing factors derived from both optical and radar satellite data for landslide susceptibility analysis. Furthermore, significant landslide predisposing factors like the soil moisture content, surface roughness, and forest biomass will be derived from radar images, and the impacts of these factors on landslide susceptibility will be examined. Hence, this study aims to investigate the efficacy of using Radar Induced Factors (RIF) for landslide susceptibility analysis under bivariate and multivariate nature.

## 1.1 Statistical Methods for Landslide Susceptibility Analysis

There are inherent limitations and uncertainties in landslide susceptibility analysis, and yet, several methods have been utilized and successfully applied in the past (Kanungo et al. 2009). These methods employed have been of both qualitative and





quantitative nature. Generally, qualitative methods are based on expert opinions while the quantitative approaches, such as statistical and probabilistic approaches, depend on the past landslide experiences.

Qualitative methods simply make use of landslide inventories to identify areas with similar geological and geomorphologic properties that show susceptibility for land failures. These methods can be divided into two groups as geomorphologic analysis, and map combination. In geomorphologic analysis, the landslide susceptibility is determined directly either in the field or by the interpretation of images through geomorphologic analysis (Bui et al. 2011). Map combination is based on combining a number of predisposing factor maps for landslide susceptibility analysis. However, map combination analysis comprises of a semi-quantitative nature by integrating the ranking and weighting of landslide susceptibility (Ayalew et al. 2004, Kavzoglu et al. 2014, Saaty 1980). The analyses based on the quantitative approaches depend on numerical data and statistics, expressing the relationship between instability or predisposing factors with landslides (Reis et al. 2012). These methods are categorized into two groups as bivariate and multivariate statistical analysis. Within the context of this work, popular Information Value method (InfoVal) as bivariate and Multi-Criteria Decision Analysis (MCDA) based on Analytic Hierarchy Process (AHP) as multivariate methods are compared with respect to their performances in landslide susceptibility modelling.

The InfoVal method determines the susceptibility at each point or pixel, jointly considering the weight of influence of all predisposing factors. The weight of influence is based on the landslide inventory map of the particular area. When constructing a probability model for landslide prediction, it is necessary to assume that the landslide occurrence is determined by landslide-related factors, and that future landslides will also occur under the same, or almost similar, conditions as past landslides (Remondo et al. 2013, Saha et al. 2005). Hence, at the beginning of the analysis, the landslide inventory map is divided in to two samples as training and validation, enabling the use of this data for landslide susceptibility analysis and validation of results respectively. The Log function is used to control the large variation of weights in calculations. Larger the weight of influence, the stronger the relationship between landslide occurrence and the given factor's attribute.

The MCDA method integrates all the independent predisposing factors with the inclusion of relative contribution of each factor by putting more emphasis on the predisposing factors that contribute to landslide occurrence. The same predisposing factors without or with radar, are used to investigate the landslide susceptibility regions from AHP technique within the GIS domain. In AHP, each pair of factors in a particular factor group is examined at one time, in terms of their relative importance. Relative weights for each factor are calculated based on a questionnaire survey from experts in the field. However, expert knowledge could be subjective at times, or may cause to assign different weights for each factor, when dealing with a large number of causative factors. Hence, in order to avoid this inconsistency, Consistency Ratio (CR) is calculated. For better predictive models, the CR should be less than 0.01, else each factor has to be generated with the proper pairwise comparison.



## 1.2 Landslide predisposing factors

It is understood that landslides may occur as consequences of complex predisposing and triggering factors. Topographical and geological factors, together with local climatic conditions, lead to landslide occurrences. The selection of these factors, and preparation of corresponding thematic data layers, are vital for models used in landslide susceptibility analysis (Jakob et al. 2006, Lee et al. 2017). There are no universal guidelines regarding the selection of predisposing factors in landslide susceptibility analysis. Some parameters may be important factors for landslide occurrences in a certain area but not for another one. Scientists (van Westen 1997, van Westen and Getahun 2003, van Westen et al. 2003) show that every study area has its own particular set of predisposal factors which condition landslides. Determination of appropriate causal factors is a difficult task, and no specific rule exists to define how many factors are sufficient for a specific landslide susceptibility analysis. Hence, the selection of predisposing factors are dependent on the nature of the study area, opinions of the experts, and the availability of data for generating the appropriate spatial and thematic information (Kavzoglu et al. 2015, Shahabi and Hashim 2015).

## 2 Study Area

Koslanda in Sri Lanka is located at the geographical coordinates of 06° 44' 00" North and 81° 01' 00" East, and the elevation is around 700 - 1000 m from the Mean Sea Level (MSL). It is a remote, hilly area with harsh weather conditions, where the monthly rainfall ranges from 60 mm to 200 mm, and average temperature is $20^0$ C. The area has rains for most of the year, with very short, dry period during the months of February to April. The population is around 5000 people, and the study area has an extent of 19 km$^2$ within the Koslanda area. Koslanda has been the site for of several massive landslides over the years, and both the Naketiya landslide in the year 1997, and Meeriyabedda landslide in the year 2014, are very distinct in Fig. 1, and within a span of two years, major landslides have occurred three times at the same location (NBRO 2016).

The geomorphology of the area is described as a gently inclined talus slope, with a thick, loosely compacted colluvium deposit at the foot of the near vertical rocky scarp. Koslanda is situated at the middle part of the slope, with the lower area showing a fairly steep surface as well. The composition of the colluvium deposit in the area includes a randomly arranged mixture of weathered clayey and sandy materials, with the organic matter making the deposit act as a sponge with high water content. The study area was an abandoned tea land in which the properly maintained surface drainage system has been neglected"(Somaratne 2016).

**FIGURE 1**





## 3 Data and methodology

The most important phases in landslide prediction analyses are the collection of data from different sources, and the construction of a spatial database for these data on a common platform (Lan et al. 2004). The data utilized for the landslide prediction analysis include the topographical, hydrological, geological, soil, and land cover factors. All factors are derived

from optical images (Landsat-8, Sentinel-2), radar images (Sentinel-1, TerraSAR-X), Digital Elevation Model (DEM) derived from aerial triangulation and other available data sources (geology, rainfall). Stereo aerial photographs from 1993 are used to generate the DEM using aerial triangulation. An inventory map of landslides for the study area was constructed by integrating the interpreted multi-temporal aerial photographs, satellite images, and some temporal images from the Google Earth. Verifications are carried out through field investigations. In this research, the predisposing factors were selected from among

the most widely considered factors in literature and opinion from the experts as depicted in the Table 1.

Most data are derived as primary data from remote sensing techniques for a large area with up-to-date information. As such, fifteen predisposing factors are selected for the landslide susceptibility analysis by using bivariate and multivariate statistical techniques. Of these, twelve factors are derived from optical images, DEM and auxiliary data, while three more factors are

derived from radar images. These factors were then combined in order to analyse the performance of this integration for landslide susceptibility analysis.

**TABLE 1**

### 3.1 Topographical Factors

The topographical factors include elevation, slope, aspect, planar curvature, profile curvature and surface roughness of the terrain. The elevation is important to study the local relief of the terrain and ranges from 446 -1537m above MSL in the study area. Since the area contains high mountains, more than a 1000m difference in elevation can be observed. The basic parameter for the slope stability analysis is the slope angle. The slope angle of the study area ranges from $0^0$ to $80^0$ degrees, showing a

significant increase of slope within a relatively small area. Additionally, the area with steep slopes ranging from $60^0$ - $80^0$ can be seen in the northern part of Koslanda. Aspect is defined as the direction of maximum slope of the terrain surface, or the compass direction of a particular slope. The curvature is theoretically defined as the rate of change of slope (or slope), of the focused slope. Planar curvature describes convergence and divergence of the flow across a surface, while the profile curvature refers to acceleration or deceleration of the flow across a surface.


Under radar configuration, the magnitude of radar backscatter is defined as a function of surface roughness and moisture content. Similar studies from Rahman et al. (2008) and Septiadi and Nasution (2009) emphasized the extraction of surface



roughness from radar data using textural analysis. Hence, to estimate the surface roughness without the use of any ancillary field data, a radar image on 12[th] March 2015 under the dry climatic condition was used to reduce the effect of the moisture component from the radar backscatter. The texture is the structure, or appearance, of the surface, and as such, describes the coarseness or the homogeneity of the image structure. One of the most prominent methods for texture analysis is Grey Level

Co-occurrence Matrix (GLCM), which is based on the second order probability density function. Hence, the GLCM texture analysis is performed using a window size of 9*9 pixels and the homogeneity or dissimilarity criterion is used to determine the surface roughness of the study area.

### 3.2    Hydrological Factors

Distance to hydrological features, rainfall, and TWI defined by Eq. (1) are selected as the hydrological factors for this landslide susceptibility analysis. Proximity to the hydrological features is an important factor when considering the landslide susceptible analyses (Sar et al. 2016, Shahabi and Hashim 2015). TWI is a solid index that is capable of predicting areas susceptible to saturation or wetness of land surfaces, and the areas that have the potential to produce an overland flow. Within the Sri Lankan context, heavy and prolonged rainfall is the main triggering factor for the landslides. The monthly average rainfall data for the

years 2014 to 2016 from 10 nearby stations to Koslanda were used in this study. Monthly rainfall data from 10 rain gauge stations are averaged, and the average rainfall map for the study area is generated using the Inverse Distance Weighting (IDW) interpolation method within the ArcGIS environment. TWI has been used to study the spatial scale effects, or topographic control, on hydrological processes. This index was developed by Beven and Kirkby (1979) and can be defined in Eq. (1) as;

$$\text{TWI} = \ln[\propto/\tan\beta] \qquad\qquad (1)$$

where $\propto$ is the local upslope area draining through a certain point per unit of contour length, and $\beta$ is the gradient of the local

slope in degrees. The applicability of the TWI in the calculation and validation of landslide susceptibility analysis has been shown by Kavzoglu et al.(2014) and Sørensen et al. (2006), among others.

### 3.3    Soil Factors

The Soil Moisture Index (SMI) defined in Eq. (2) and Delta Index defined in Eq. (5) are the soil factors focused upon in this

research. Surface soil moisture is one of the most important parameters in land susceptibility analysis (Carlson et al. 1994, Zhan et al. 2002). Several methods have been proposed to estimate the surface soil moisture conditions accurately with in situ



measurements. However, these methods are time consuming and costly when the area of interest is large, and the scale of work is small. Hence, this research uses the Universal Triangle relationship between Soil Moisture, Normalized Difference Vegetation Index (NDVI) and Land Surface Temperature (LST) derived from Landsat-8 image bands as an optical remote sensing approach, and the Delta Index derived from two radar images, as wet and dry conditions, as a radar remote sensing

5  approach. Band 5 (Near Infrared (NIR), 30m resolution), band 4 (Red, 30m resolution) and band 11 (Thermal, TIR-2, 100m resolution) of Landsat-8 image of 3rd July 2015 is processed for extracting the soil moisture index in the Thermal-NDVI space. The SMI is "0" along the dry edge and "1" along the wet edge. According to the studies from (Wang and Qu 2009, Zenga et al. 2004), SMI can be defined in Eq. (2) as;

$$SMI = \frac{(T_{max} - T)}{(T_{max} - T_{min})} \qquad (2)$$

where $T_{max}$, $T_{min}$ are the maximum and minimum surface temperature for a given NDVI, and T is the remotely sensed derived surface temperature at a given pixel for a given NDVI. The simple regression relationship between T and NDVI is formulated in Eq. (3) and Eq. (4) as;

$$T_{max} = a_1 \cdot NDVI + b_1 \qquad (3)$$
$$T_{min} = a_2 \cdot NDVI + b_2 \qquad (4)$$

where, $a_1$ = -5.2362, $b_1$ = 300.14, $a_2$ = 2.9254, and $b_2$ = 289.11.

Radar remote sensing provides advantages for extracting near surface soil moisture (0-5cm), including timely coverage with repeat passes during day and night, under all weather conditions. Technically, the surface roughness and vegetation affect radar backscatter much more than soil moisture. Hence, both the surface roughness and vegetation have to remain unchanged during the image acquisition for soil moisture estimation (Thoma et al. 2006). Delta Index is a modified, image differencing

20  technique, and many studies (Barrett et al. 2009, Sano et al. 1998, Thoma et al. 2004) have proven it to be a good predictor for near surface soil moisture extraction. This index describes the change of wet scene backscatter relative to the dry scene backscatter, and is defined by Thoma et al.(2004) in Eq. (5) as;

$$Delta\ Index = \left| \frac{\sigma^0_{wet} - \sigma^0_{dry}}{\sigma^0_{dry}} \right| \qquad (5)$$

25  where, $\sigma^0_{wet}$ is the radar backscatter (decibels) from a pixel in the radar image representing wet soil conditions, and $\sigma^0_{dry}$ is the radar backscatter (decibels) from a pixel in the same geographic location representing dry soil conditions at a different time. Sentinel-1 images with 10m spatial resolution and VV polarization is used in the presented study. The dry reference image was acquired on 12th March 2015 and the wet image was acquired on 24th November 2014 after the landslide in Meeriyabedda,



Sri Lanka. Therefore, the topographical changes like roughness and vegetation density showed no significant changes during these four months' time.

### 3.4 Land Use

The major land uses existing in this study area are identified as tea, scrub, forest, rock, rice, water, and residential. The Sentinel-2A image from 10th October 2016 is used to extract the desired land uses from the study area by applying supervised classification. Scrub areas are typically the tea estates that are in abundance, while the residential areas are the rooms of tea workers. It is noted that most of the devastating landslides in this area had occurred within the extensive tea estates. Hence, the main reason for the continuous occurrence of these landslides can be identified as the lack of proper land use management in the area.

Forest biomass is a significant factor that can control the landmass failures or landslides. The main limitations of using optical remote sensing for forest biomass estimation is the near constant tropical cloud cover, and the insensitivity of reflectance to change of the biomass in older and mixed forests. Radar has potential to overcome the above limitations due to its all-weather, day and night capability, with the positive relationship of radar backscatter and forest biomass. Kuplich et al. (2005) and Caicoya et al.(2016) related the radar image texture derived from GLCM to the forest biomass. An experiment was conducted by Kuplich et al. (2005) with seven texture measures, but only the GLCM derived contrast increased the correlation between the backscatter and the log of biomass in Eq. (6) as;

$$Log\ of\ Biomass = 2.24 + 0.33b + 0.0001c \qquad (6)$$

where, b is the radar back scatter and the c is the GLCM contrast texture for the particular radar image. TerraSAR-X spot light image from 2nd November 2014, with 3m resolution and dual polarization (HH and VV), was used to estimate the forest biomass in this work.

### 3.5 Geological Factors

Geology refers to the physical structure and the substance of the Earth. In order to investigate the land mass failures, the geological structure of that particular area have to be analysed carefully. In addition to the Geology of the area, lineament density has also been considered as a factor. The geological information of the particular area is obtained from the geological map available at the Geological Survey Mines Bureau (GSMB), Sri Lanka at 1:100000 scale, and seven types of different geological structures are contained in the selected study region. Lineaments are extractable linear features which are correlated with the geological structures of the earth. When considering the analysis of lineaments with respect to the landslide





potentiality, lineaments exhibit the zones of weakness surfaces as faults, fractures, and joints (Mandal and Maiti 2015). This study uses the Sentinel-2A optical satellite image, with 10m resolution, for the extraction of lineaments of the study area.

After decisive analysis of the types of predisposing factors, the presented work proceeded to consider fifteen predisposing factors that are derived from optical, radar and other available auxiliary data sources. Three significant causative factors as surface roughness, soil moisture from Delta Index, and forest biomass were estimated by using radar satellite images. Thus, this work investigated the performance of landslide susceptibility analysis using bivariate and multivariate methods with the inclusion of RIF and described the processing steps in Fig. 2.

The weight of influence of all predisposing factors as thematic maps are added in bivariate and multivariate nature to obtain the contribution of all predisposing factors for landslide susceptibility analysis. After calculating the cumulative percentage of failures of the weighted susceptibility maps, value ranges for each percentage of failure are obtained from quantile classification for 10 classes. The entire study area of each landslide susceptibility map is then discretized in to four classes as 0%, 10%, 30% and 60% of failure regions for very low, low, moderate, and high susceptibility classes, respectively.

**FIGURE 2**

## 4    Results and Discussions

Four Landslide prediction models, (i) bivariate without RIF (BiNR), (ii) bivariate with RIF (BiWR), (iii) multivariate without RIF (MNR), and (iv) multivariate with RIF (MWR) are discussed. The region has been analysed and classified into four (04) landslide susceptibility regions as; high, moderate, low, and very low.

### 4.1    Bivariate InfoVal method Without and With RIF

Susceptible regions are identified from the bivariate InfoVal method without RIF as 12% for high, 45% for moderate, 38% for low, and 5% for very low as shown in Fig. 3 (a). Hence, 57% areas from the total study area are predicted as having high and moderate susceptibility for the landslide hazard. Very steep slope mountains in the North, North West, and East regions are identified as very low susceptibility areas, given that the area was free from historical landslides. The middle regions with $30^0$-$50^0$ slope are detected as having a high probability for landslide occurrences. The bivariate InfoVal method with RIF identified 19% of failure regions for high susceptibility, 39% for moderate, 33% for low, and 9% for very low susceptible regions as presented in Fig. 3 (b). Therefore, 58% of the total study area is predicted as having high and moderate susceptibility for landslides. Very steep slope mountains in the North, North West, East, and South East regions, the area near the Eruwendumpola Oya, are identified as having very low susceptibility for landslides. Similar to the bivariate analysis without



RIF, the middle regions with $30^0$-$50^0$ slope are detected as having high probability for landslide occurrences and the reason for this is mainly with the past experience from Naketiya and Meeriyabedda landslides that had taken place in the same area. As mentioned before in Sec. 1.1, the presented work utilizes the historical database of landslides and the land failures in the same region.

## 4.2 Multivariate MCDA based on AHP Without and With RIF

All fifteen weighted predisposing factors were grouped as without and with RIF, and weighted overlay is performed separately in order to obtain the landslide susceptibility regions. The calculated weights for elevation, slope, aspect, planar curvature, profile curvature, TWI, land use, lineament density, distance to water bodies, SMI in NDVI-T domain, geology, and rainfall

are 0.030, 0.172, 0.022, 0.018, 0.014, 0.074, 0.149, 0.052, 0.045, 0.094, 0.185, and 0.145, respectively. The Consistency Ratio (CR) is a measure of consistency in subjective judgement, and ranges from 0 to 0.1, where 0 indicate the maximum inconsistency of relative judgement and 0.1 indicate the maximum consistency of relative judgements. For the present work, the CR for the relative judgement of weighting predisposing factors is 0.089 in the pairwise comparison. The weights for the fifteen predisposing factors with RIF, as elevation, slope, aspect, planar curvature, profile curvature, TWI, land use, lineament

density, distance to water bodies, SMI in NDVI-T domain, geology, rainfall, soil moisture (Delta index), surface roughness, and forest biomass are 0.022, 0.145, 0.016, 0.013, 0.011, 0.053, 0.126, 0.039, 0.033, 0.065, 0.153, 0.124, 0.088, 0.088, and 0.027, respectively. When considering the fifteen predisposing factors, the CR is 0.092, which is less than the 0.1 thereby showing a realistic level of consistency in the pairwise comparison matrix.

**FIGURE 3**

Figure 3 (c) illustrates the landslide susceptibility map from the multivariate method without RIF and is able to identify 18% for high, 44% for moderate, 36% for low and 2% for very low susceptible regions. Hence, 62% of areas from the total study area are predicted to be of high and moderate susceptibility for the landslide hazard. In the landslide susceptibility map from

the multivariate method with RIF, from the total area, 21% of the area show a high susceptibility to landslides, with 40% of area as moderate, 34% area as low, and 5% of area as having very low susceptibility as shown in Fig. 3 (d). Hence, 61% of areas from the study area are predicted as having high and moderate susceptibility for the landslide hazard. In a similar manner to the InfoVal method, the top of the mountains in the North, North West, East, and South East regions, area near to the Eruwendumpola Oya, are identified as having a very low susceptibility to landslide hazards, while the middle regions with

$30^0$-$50^0$ slopes are detected as having high and moderate probability for landslide occurrences.

The area identified as having high and moderate susceptibility classes in these four approaches (57%, 58%, 62%, and 61% respectively in BiNR, BiWR, MNR, and MWR) are close in value, but shows an increase in multivariate analysis when





compared with bivariate analysis as tabulated in Table 2. Moderate and low landslide susceptibility areas show very small ((1-2) %) changes between these four types of analysis. With the integration of RIF as surface roughness, near surface soil moisture from Delta Index, and forest biomass in bivariate and multivariate analysis, the high and very low susceptible areas are increased significantly (high: 7% - bivariate, 3% - multivariate and very low: 4% - bivariate, 3% - multivariate). However,

when comparing the high and very low susceptibility areas from bivariate and multivariate analysis, high susceptibility areas show a considerable increase (without radar: 6% and with radar: 2%) while, very low susceptibility areas have a noteworthy decrease (without radar: 3% and with radar: 4%).

**TABLE 2**

## 4.3     Results Validation

The landslide susceptibility maps derived from the bivariate and multivariate analysis are validated using the selected validation samples from the landslide failure map. The most commonly used and scientifically recognized Receiver Operating Characteristics (ROC) curves are used to analyse the prediction and validation performances. ROC is a graphical plot that

illustrates the performance of classification, and is considered as a powerful tool for the validation of landslide susceptibility analysis for many years (Neuhäuser et al. 2012). The Area Under Curves (AUC) for the four different approaches, as bivariate and multivariate without and with RIF, are calculated and graphed in Fig. 4.

**FIGURE 4**

The areas under the success rate curves measure how the landslide prediction analysis fit with the training data set, while the areas under the prediction rate curves measure how well the landslide prediction models and landslide causative factors predict the landslides. If the area under the ROC curve is closer to 1, the result of the test is excellent and vice versa, and when AUC is closer to 0.5, the result of the test is fair or acceptable (Kamp et al. 2008).


The AUC of all the success rates are more-or-less near 0.80, indicating good prediction performances according to the definition. The AUC of all the prediction rates are having values above 0.50, thereby indicating that they are within the acceptable range as per the definition. As such, they indicate that the accuracy of prediction rate of land susceptibility and the selection of land causative factors are acceptable, but not excellent, even though the fit between the landslide prediction and

the training data set are excellent as compared in Table 3. The incompleteness of the available landslide inventory map, as well as an insufficient number of validation samples in the study area can be shown as reasons for the discrepancy. As a whole,



better prediction and validation capabilities are shown by the bivariate analysis when compared with the multivariate approaches.

**TABLE 3**

**5 Conclusions**

The main difference between bivariate and multivariate analysis is that in multivariate analysis, selected predisposing factors are also weighted by considering how each of them influence landslides. This study investigated fifteen landslide predisposing factors as elevation, slope, aspect, planar curvature, profile curvature, TWI, land use, lineament density, distance to hydrology, SMI in NDVI-T domain, geology, rainfall, soil moisture (Delta Index), surface roughness, and forest biomass. Most of the

factors are derived from radar and optical remote sensing techniques, where smaller scale studies with up-to-date information allows the work to be conducted at the meter-level accuracy, and repeated analysis simultaneously.

From the results obtained, it can be concluded that the bivariate and multivariate statistical analysis, without and with RIF, can be used for landslide prediction analysis. However, with the integration of RIF as surface roughness, near surface soil moisture

from Delta Index, and forest biomass, the detection of the boundary between the high and very low susceptibility areas is increased. When comparing the bivariate analysis with the multivariate analysis, the area identified as high susceptibility regions are increased while very low susceptibility regions decreased. As a whole, there is an improvement of prediction and validation performances of bivariate analysis than multivariate analysis.

This study focused on the applicability of remote sensing and GIS for rapid landslide prediction analysis at finer scale. Further, by considering the significance of radar data for landslide analysis, this study mainly investigates the efficacy of radar induced factors for landslide prediction analysis which is not well experimented in the current researches. Most significant factors as surface roughness, soil moisture, and forest biomass derived from radar are incorporated to examine the landslide prediction analysis. Successful prediction and validation of prediction analysis via ROC curves are achieved. Even though this study was

tested for a sample area, the same methodology can be applied for any landslide prone area to investigate the landslide prediction analysis using radar induced factors by using bivariate and multivariate analysis. This is because the radar induced factors can be derived for any area, as long as the data are available, and at any time under whatever the weather conditions as radar are weather independent. Additionally, the technology can be learned easily and anyone can be trained to use this methodology to predict landslide susceptibility areas, and this is especially helpful for developing countries who do not have

up-to-date data at fine resolutions.



With the increasing availability of free data in optical, radar, and DEM, it is possible to derive more landslide predisposing factors as thematic maps. Further, there are many statistical analyses developed in qualitative and quantitative natures for spatial data analysis. Hence, further investigations will result in landslide susceptibility analysis even focusing the changing nature of the environments.

**Team List**

A.K.R.N. Ranasinghe
R. Bandara
U.G.A. Puswewala
T.L. Dammalage

**Author Contribution**

A.K.R.N. Ranasinghe performed the Conceptualization, Data curation, Formal analysis, Funding acquisition, Investigation, Methodology, Validation, Visualization, and Writing – original draft and R. Bandara accomplished the Supervision, and
writing - review and editing. U.G.A. Puswewala and T.L. Dammalage executed the Supervision.

**Competing Interests**

The authors declare that we have no conflict of interest.

**Acknowledgement**

Authors wish to acknowledge the Sabaragamuwa University of Sri Lanka for offering an opportunity for this research and the HETC project, Ministry of Higher Education, Sri Lanka, for providing financial support under the grant number SUSL/O-Geo
/N2 and the University of Siegen, Germany, for providing their support in collecting and initial processing of the TerraSAR-X images from DLR, Germany. The DLR, Germany, is remembered with appreciation for providing radar images for free of charge, and the GSMB, Sri Lanka for providing freely, the geological data necessary for this research work.

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



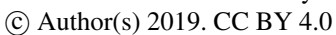


**Figure 1 Topographical formation of Koslanda, Sri Lanka with its previous Landslides Signatures**





Figure 2 Methodological flow of the Landslide susceptibility analysis using Bivariate and Multivariate approaches





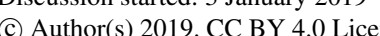

**Figure 3 Landslide susceptibility maps from bivariate and multivariate analysis without and with RIF. (a)- bivariate without RIF, (b)- bivariate with RIF, (c)- multivariate without RIF, and (d)- multivariate with RIF**





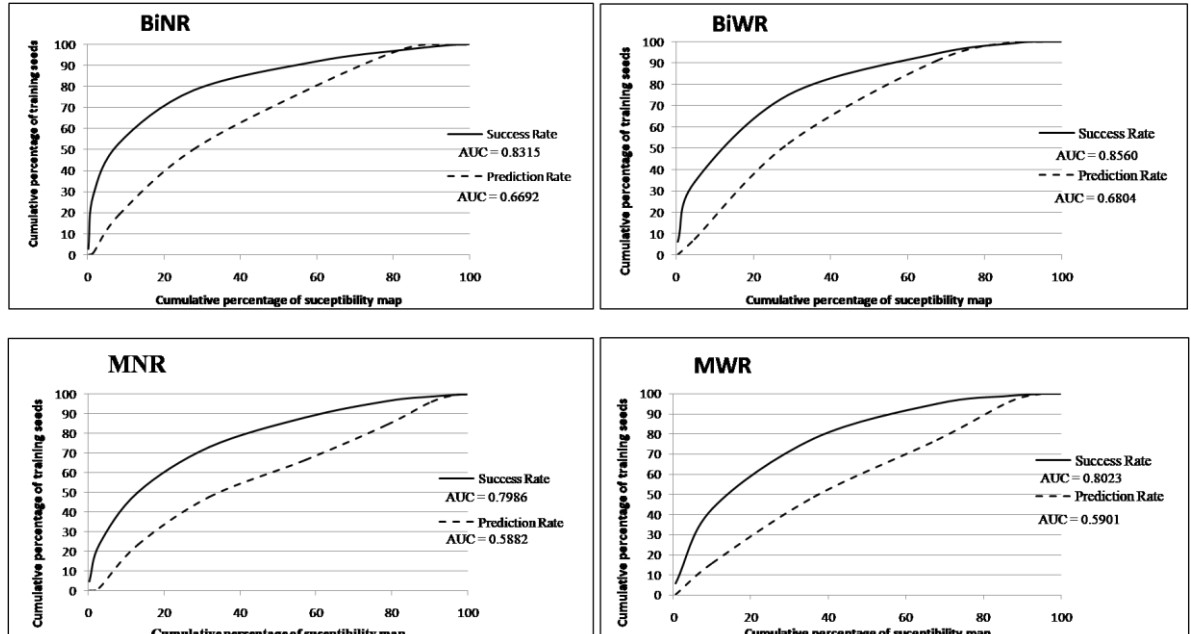

**Figure 4 Success rate and Prediction rate curves with AUC for the bivariate and multivariate analysis without and with RIF. X axis denotes the Cumulative percentage of Susceptibility regions and Y axis denotes the Cumulative percentage of training samples. From left to right and top to bottom BiNR- bivariate analysis without RIF, BiWR- with RIF, and MNR- multivariate analysis without RIF, and MWR- with RIF**





**Table 1 Selected Predisposing Factors for Landslide susceptibility analysis**

| Factors | |
|---|---|
| Main Factors | Sub Factors |
| Topographical | Elevation |
| | Slope |
| | Aspect |
| | Planar Curvature |
| | Profile Curvature |
| | Surface Roughness (*radar*) |
| Hydrological | Distance to Water Bodies |
| | TWI |
| | Rainfall |
| Soil | Surface Soil Moisture |
| | Soil Moisture Index (*radar*) |
| Land cover | Land Cover Type |
| | Forest Biomass (*radar*) |
| Geological | Geology |
| | Lineament |



**Table 2 Landslide susceptible area comparison from bivariate and multivariate analysis without and with RIF, BiNR - Bivariate analysis without RIF, BiWR - Bivariate analysis with RIF, MNR - Multivariate analysis without RIF, MWR - Multivariate analysis with RIF**

|  | BiNR | BiWR | MNR | MWR |
|---|---|---|---|---|
| High | 12% | 19% | 18% | 21% |
| Moderate | 45% | 39% | 44% | 40% |
| Low | 38% | 33% | 36% | 34% |
| Very Low | 05% | 09% | 02% | 05% |



**Table 3 Comparison of area under Success rate and Prediction rate curves for bivariate analysis without RIF (BiNR), with RIF (BiWR), and multivariate analysis without RIF (MNR), and without RIF (MWR).**

| AUC | BiNR | BiWR | MNR | MWR |
|---|---|---|---|---|
| Success rate | 0.8315 | 0.8560 | 0.7986 | 0.8023 |
| Prediction rate | 0.6692 | 0.6804 | 0.5882 | 0.5901 |