# Peer review of "Efficacy of using Radar Derived Factors in Landslide Susceptibility Analysis: case study of Koslanda, Sri Lanka."

_Natural Hazards and Earth System Sciences, 2018_

## Referee Comment (RC1) · Anonymous Referee #1 · 22 Jan 2019

Dear authors, the manuscript titled "Efficiency of using Radar Induced Factors in landslide susceptibility analysis: case study of Koslanda, Sri Lanka" deals with the application of 4 methods for assessing the landslide susceptibility map: Bivariate InfoVal, Bivariate InfoVal with RIF, Multivariate MCDA based on AHP, Multivariate MCDA based on AHP with RIF. The work generally fits the aim of the journal, but needs several modifications and some lacks needs to be filled. The required improvements interest several parts of the work, mainly the data and methodology, for this reason I suggest major revision.

[Figure]

General Comments First of all, why all the images are black and with? Please provide the coloured images. The Abstract has to be rewritten because it cannot stay alone to explain the conducted work. The Introduction, besides it is quite long, partially misses in state-of-the-art about the landslide susceptibility methods and, mainly, in the use of remote sensing data for susceptibility map purposes and for extracting the parameters then utilized. The study area is not welle presented. It is not well localized (also because the images are B/W) and described. Furthermore, no geological and geomorphological information of the area were inserted. These information have to be added. Data and methodology section has to be deeply improved. Add more info and images of the used data, while Table 1 can be removed because it is useless and it no add information with respect to the text. The used methodologies are no described, as well as no images of the described and used factors are present. How did you extract factors by Sentinel-2 and Landsat images? The resolution of the images was enough? Please clarify. Insert the landslide inventory map derived by the multi-temporal analysis. Table 1 can be maintained if the relative weights are included, with a short explanation about how they were calculated and the addition of the "questionnaire survey form" in the text or as supplementary material. Results and discussion also need improvements. I suggest to separate the results and discussions. In the results session the four resulting landslide susceptible maps calculated (please with colours) have to be inserted and described. To make readable and comparable all the percentage of the four maps and relative classes, please summarize them in a table. Then the comparison between then can be insert in the discussions session. Consequently to all the required modifications and suggestions the conclusion has to be reviewed accordingly.

Minor issues - Add some more recent references about the "landslide-specific information for emergency and disaster management activities in the world". See for example Solari et al., 2018 - Add references of already published methods, e.g. IDW, NDVI and LST - Pay attention to the tenses. You write some parts using the present form and other the past. Please check - Line24 page 2 - remove "could" - Substitute "from the Mean Sea Level" with a.s.l. (above sea level) - Line26 page 4 - remove " - Line 3 page

5 remove "for these data" - Line7 page 5 - Substitute "an inventory map of landslide" with "landslide inventory map"

---

## Short Comment (SC1) · 22 Jan 2019

The manuscript shows the comparison among different approaches (bivari-ate/multivariate analyses) using different sets of data (classic/classic + radar data) to produce a landslide susceptibility map of an area located in Sri Lanka. The work in general seems appropriate for the journal but it is not very well organized. In the paper a reader would expect to read: 1) a comprehensive introduction with proper literature, 2) a detailed description of the study area and its problematic in terms natural hazard; 3) a description of the adopted methodology; 4) the presentation of the results, 5) a

discussion of the obtained results; 6) final remarks. I think the manuscript contains some of these issues but not well organized.

The introduction session is very long with respect to the rest of the paper. The authors should add some background knowledge about the use of remote sensing data and in particular of radar data to infer topographical, soil and land cover information. The literature review part in the first part of the Introduction needs to be improved. The second part (Statistical methods for landslide susceptibility analysis) should be reduced and part of it should be moved into the methodology description. The description of the study area is very short. Please add some information about the geology of the study area and about the typology of the landslides which affect the study area. The section "Data and methodology" is actually a list of the data available. There is nothing about the bivariate or multivariate methods. I suggest to show a map for each considered predisposing factor. Some factors need for a more accurate description, for example you need to describe the geology of the study area (Geological factors), in this paragraph information about the geology of the study area and the used classes totally lack. How do you decided the weight of influence of all predisposing factors? I suggest to split the results from the discussion. In the results section you need to present the landslide susceptibility maps and to explain their significance in terms of predisposing factors. In the discussion you can compare all the obtained maps highlighting advantages, drawbacks and limitation. Figure 1: I think that a colour figure can have more appeal, the same for figure 3. Minor issue: Page 1 Line 23: I think that you mean 90% and not 09% Page 2 Line 11: Earth and not earth Page 2 Line 33: delete "employed" Page 4 Line 24: "act as a sponge" does not sound really scientific Page 5 line 5: how much is the DEM resolution? Page 7 line 6: what does "Thermal-NDVI space" mean? Page 9 Line 2: How do you extracted the lineaments from Landsat and Sentinel 2 images? Are you sure that joints and fractures can be observed with the resolution of Landsat and Sentinel? Several references are not reported in the reference list: (van Vesten 1997; Somaratne, 2016; Rahman et al., 2008; Septiadi and Nasution 2009; Zhan et al., 2002)

I suggest to reject the paper.

---

## Referee Comment (RC2) · Anonymous Referee #2 · 8 Mar 2019

The paper deals with a topic of interest for the journal. I think it could be of interest for the readers. However, in my opinion there is still work to be done in order to make it suitable for publication. Here below my comments: - As the other reviewers, I think it is not well organized. The abstract is "strange". It is not a good summary of the paper. I think also that the introduction is not well focused and too long. And I see too long sentences which sometimes makes difficult the understanding. Can you improve it?

- The quality of the figures is poor. Why don not use colour figures??

- The analysis of the results is also very qualitative. In the conclusion, the authors say that "with the integration of RIF as surface roughness, near surface soil moisture 15 from Delta Index, and forest biomass, the detection of the boundary between the high and very low susceptibility areas is increased". However, it is not well demonstrated from the given results and explanation. Can you improve it?

- I am not sure that from the result one can conclude that RIF helps to improve the results. I see very similar results by using and by not using the RIF parameters. Please, can you improve your analysis in order to be more convincent or change the conclusion?

Nicel luck and best regards

---

## Author Comment (AC1) · 4 Apr 2019

Dear Editor in Chief

We are pleased to submit a revised manuscript entitled Efficacy of using Radar Induced Factors in Landslide Susceptibility Analysis: case study of Koslanda, Sri Lanka for publication in the Journal of Natural Hazards and Earth System Sciences. A revised copy of the manuscript is provided with changes to the manuscript requested by the reviewers indicated in the attached document, together with detailed responses to the

reviewers' comments.

Yours Sincerely AKRN Ranasinghe

Please also note the supplement to this comment:
https://www.nat-hazards-earth-syst-sci-discuss.net/nhess-2018-335/nhess-2018-335-AC1-supplement.zip
* * *

---

## Author Response (AR1)

Dear Editor in Chief

We are pleased to submit a revised manuscript entitled *Efficacy of using Radar Derived Factors in Landslide Susceptibility Analysis: case study of Koslanda, Sri Lanka* for publication in the Journal of Natural Hazards and Earth System Sciences. A revised copy of the manuscript is provided with changes to the manuscript requested by the editor and reviewers indicated in the attached document, together with detailed responses to the editor's and reviewers' comments.

Yours Sincerely

AKRN Ranasinghe

**Responses to editor and reviewer comments on the paper " Efficacy of using Radar Derived Factors in Landslide Susceptibility Analysis: case study of Koslanda, Sri Lanka"**

We wish to thank the editor and all reviewers for their constructive comments. Both the editor and the reviewers felt that the paper has to be well organized and the introduction part should be reduced by moving some parts to the methodology. Specifically, the description of the study area should be made a little larger by adding information about the geology and the typology of the landslides. Further, they have commented on the rewriting of the abstract and the conclusions according to the conducted research work. All reviewers stated on the inclusion of colour figures as they are more visually appealing. Consequently, the paper has been rearranged, the abstract rewritten, introduction reduced, methodology rearranged, and the study area expanded. Additionally, the Results and Discussion were separated and Conclusions changed accordingly. All the figures were inserted in colour by preserving the colour blindness using the colour scales. The details of these changes are provided below, along with the responses to the comments.

In the following, the comments of the editor and reviewers are shown in italics and our responses indented in normal text. References to the edited lines are according to those found in the revised manuscript, unless specifically referred to in the original manuscript.

**Response to Editor**

*After carefully revising your paper and the discussion, we believe that your manuscript is of potential interest to the journal but not ready for publication. A significant improvement could be reached by introducing major changes which, according to the referee's reports, should mainly focus on:*

1) *Modifying the paper structure by reducing the introduction, increasing the geological and geomorphological description of the test site, separating results from the discussion and modifying the methods section, as well.*

The introduction part has been improved by reducing the extra information where unnecessary. The following lines have been deleted; page 1 line 29, lines 32-34, page 2 lines 1-3, line 13, and lines 19-20.

Furthermore, part of the statistical methods for landslide susceptibility analysis was moved to the methodology part. (page 2 lines 41-42 and page 3 lines 1-17 to page 7 lines 35-39, page 8 lines 1-4, lines 18-22 and lines 25-28).

Remote sensing for susceptibility map purposes and for extracting the parameters are already explained in page 2 lines 5-11.

The geomorphological information to the test site is already included in the manuscript in page 4 lines 1 – 6, but additional geological information is inserted in to page 4 lines 6-10 as;

"Geology refers to the physical structure and the substance of the Earth. The study area consists mainly of undifferentiated charnockitic biotite gneisses and Quartzites, according to the 1:10000 geological map from the Geological Survey Mines Bureau (GSMB), Sri Lanka. Such geomorphological and geological formation, together with improper land use management practices, has made the area extremely vulnerable for landslide events."

Data and Methodology sections have been separated, while improving the methodology part. (page 4 line 15, page 7 line 33 to page 9 line 11).

Table 1 is removed from the manuscript (page 4 line 35), while inserting the types of predisposing factors in to the text (page 4 lines 29-32) as;

"Of these, twelve factors (elevation, slope, aspect, planar curvature, profile curvature, Topographical Wetness Index (TWI), land use, lineament density, distance to water bodies, soil moisture, geology, and rainfall) are derived from optical images, DEM and auxiliary data, while three more factors (soil moisture from Delta Index, surface roughness, and forest biomass) are derived from radar images."

The Discussion part has been separated from the Results section. (page 9 line 14)

In the results section, four colour landslide susceptibility maps were inserted in to the manuscript. (Figure 4)

Resultant susceptibility regions as high, moderate, low and very low regions are numerically compared with the spatial formation in the study area. (page 9 lines 20-31) and (page 10 lines 12-21)

2) *Rewriting the abstract so it is self-contained and can stand alone*

The abstract has been rewritten. (page 1, lines 9 – 20)

3) *The methods used and the procedural steps are not adequately described. It seems that the approach you follow is not clear and could not be based on robust standards. Could you please explain and improve this section?*

Data and Methodology sections have been separated, while improving the methodology part. (page 4 line 15, page 7 line 33 to page 9 line 11).

Statistical analysis of bivariate and multivariate methods for landslide susceptibility analysis is inserted in to the manuscript. (page 7 lines 35-39, page 8 lines 1-4, page 8 lines 18-22, and lines 25-28).

The details of the relative weight calculation in bivariate, information value method (page 8 lines 8 – 16) and multivariate, MCDA based on AHP is inserted in to the manuscript (page 8 lines 18 – 28).

4) *Introducing some quantitative approach to validate the efficiency of your method*

All the susceptibility analysis and validations are quantitative. In susceptibility analysis, all the predisposing factors are overlaid with the training sample from landslide failure map and the weight of susceptibility index for landslide occurrences was calculated. Then, by using the bivariate and multivariate analysis, landslide prediction models are generated with and without radar derived factors. Hence, all the landslide prediction analysis is quantitative. Similarly, all the validations are performed by overlaying the validation samples on the four different landslide susceptibility maps and calculating the

Cumulative landslide seeds in training and validation samples with respect to the landslide susceptibility classes providing quantitative approach of AUC.

5) *As a side (but important) issue: I would personally advise against using the expression "radar Induced factors" both in the title and in the text. Radar methods are mainly a measuring tool and it is very unlikely that they will be able to induce any landscape factor or to modify it.*

Corrected "radar induced factors" as "radar derived factors" in the whole paper, starting from the topic as "Efficacy of using Radar Derived Factors in Landslide Susceptibility Analysis: case study of Koslanda, Sri Lanka". (page 1 line 1)

The phrase "Radar Induced Factors (RIF)" has been replaced with "Radar Derived Factors (RDF)" in the complete manuscript.

**Response to Anonymous Referee #1.**

*Dear authors, the manuscript titled "Efficiency of using Radar Induced Factors in landslide susceptibility analysis: case study of Koslanda, Sri Lanka" deals with the application of 4 methods for assessing the landslide susceptibility map: Bivariate InfoVal, Bivariate InfoVal with RIF, Multivariate MCDA based on AHP, Multivariate MCDA based on AHP with RIF. The work generally fits the aim of the journal, but needs several modifications and some lacks needs to be filled. The required improvements interest several parts of the work, mainly the data and methodology.*

*RC1-1   Why all the images are black and white? Please provide the coloured images.*

All the figures have been inserted in to the manuscript in colour using the given colour scales. (Figure 1, 2, & 4)

*RC1-2   The Abstract has to be rewritten because it cannot stay alone to explain the conducted work.*

The abstract has been rewritten. (page 1, lines 9 – 20)

*RC1-3   The Introduction, besides it is quite long, partially misses in state-of-the-art about the landslide susceptibility methods and, mainly, in the use of remote sensing data for susceptibility map purposes and for extracting the parameters then utilized.*

The introduction part has been improved by reducing the extra information where unnecessary. The following lines have been deleted; page 1 line 29, lines 32-34, page 2 lines 1-3, line 13, and lines 19-20.

Furthermore, part of the statistical methods for landslide susceptibility analysis was moved to the methodology part. (page 2 lines 41-42 and page 3 lines 1-17 to page 7 lines 35-39, page 8 lines 1-4, lines 18-22 and lines 25-28).

Remote sensing for susceptibility map purposes and for extracting the parameters are already explained in page 2 lines 5-11.

*RC1-4   The study area is not well presented. It is not well localized (also because the images are B/W) and described. Furthermore, no geological and geomorphological information of the area were inserted. These information have to be added.*

A colour image with topographical information, with previous landslide signatures, has been inserted as Figure 1.

The geomorphological information to the test site is already included in the manuscript in page 4 lines 1 – 6, but additional geological information is inserted in to page 4 lines 6-10 as;

"Geology refers to the physical structure and the substance of the Earth. The study area consists mainly of undifferentiated charnockitic biotite gneisses and Quartzites, according to the 1:10000 geological map from the Geological Survey Mines Bureau (GSMB), Sri Lanka. Such geomorphological and geological formation, together with improper land use management practices, has made the area extremely vulnerable for landslide events."

*RC1-5    Data and methodology section has to be deeply improved. Add more info and images of the used data, while Table 1 can be removed because it is useless and it no add information with respect to the text. The used methodologies are no described, as well as no images of the described and used factors are present.*

Table 1 is removed from the manuscript (page 4 line 35), while inserting the types of predisposing factors in to the text (page 4 lines 29-32) as;

"Of these, twelve factors (elevation, slope, aspect, planar curvature, profile curvature, Topographical Wetness Index (TWI), land use, lineament density, distance to water bodies, soil moisture, geology, and rainfall) are derived from optical images, DEM and auxiliary data, while three more factors (soil moisture from Delta Index, surface roughness, and forest biomass) are derived from radar images."

The information and the images used to extract all landslide predisposing factors are already within the manuscript under all predisposing factors. However, some information has been added as (page 4 line 39 – page 5 line 1)

Available information is, for example, topographical factors –"Sentinel-1 radar image on 12$^{th}$ March 2015" (page 5 line 13), Soil factors – "Landsat-8 image of 3$^{rd}$ July 2015" (page 6 line 10) and "dry reference image on 12$^{th}$ March 2015 and the wet image on 24$^{th}$ November 2014" (page 6 lines 32-33).

Data and Methodology sections have been separated, while improving the methodology part. (page 4 line 15, page 7 line 33 to page 9 line 11).

When considering the guide-lines for manuscript preparation, even though the individual figures from fifteen predisposing factors are really significant, it is difficult to add them all to the manuscript. Hence, all the fifteen predisposing factors (in colour figures) were added as supplementary materials (Sup 1-3).

*RC1-6    How did you extract factors by Sentinel-2 and Landsat images? The resolution of the images was enough? Please clarify.*

Sentinel – 2 images with 10 m resolution is used to extract Land use (page 8 lines 13 – 15) and Lineament predisposing factors. Landsat-8 with 30 m resolution (NIR & R bands) and 100m resolution Thermal band is used to extract surface soil moisture from

Universal Triangle relationship between Soil Moisture, Normalized Difference Vegetation Index (NDVI), and Land Surface Temperature (LST). The study area is approximately 19 km$^2$ and since this study is primarily focused on the applicability of remote sensing (radar and optical) for landslide susceptibility analysis on smaller scale, the freely available Sentinel–2 and Landsat–8 image with Thermal band was sufficiently enough for this research study.

*RC1-7   Insert the landslide inventory map derived by the multi-temporal analysis.*

Landslide inventory map with training and validation samples for Landslide susceptibility analysis has been inserted in to the manuscript as Figure 2. (page 8 line 6)

*RC1-8  Table 1 can be maintained if the relative weights are included, with a short explanation about how they were calculated and the addition of the "questionnaire survey form" in the text or as supplementary material.*

Table 1 is removed from the manuscript (page 4 line 35), while inserting the types of predisposing factors in to the text (page 4 lines 29-32) as;

"Of these, twelve factors (elevation, slope, aspect, planar curvature, profile curvature, Topographical Wetness Index (TWI), land use, lineament density, distance to water bodies, soil moisture, geology, and rainfall) are derived from optical images, DEM and auxiliary data, while three more factors (soil moisture from Delta Index, surface roughness, and forest biomass) are derived from radar images."

 The details of the relative weight calculation in bivariate, information value method (page 8 lines 8 – 16) and multivariate, MCDA based on AHP is inserted in to the manuscript (page 8 lines 18 – 28).

The sample questionnaire has been attached as a supplementary material (Sup 4)

*RC1-9   Results and discussion also need improvements. I suggest to separate the results and discussions. In the results session the four resulting landslide susceptible maps calculated (please with colours) have to be inserted and described.*

The Discussion part has been separated from the Results section. (page 9 line 14), Four colour landslide susceptibility maps have been inserted in to the manuscript under the results section (Figure 4)

Resultant susceptibility regions as high, moderate, low and very low regions are numerically compared with the spatial formation in the study area. (page 9 line 20-31, and page 10 lines 12-21)

*RC1-10   To make readable and comparable all the percentage of the four maps and relative classes, please summarize them in a table. Then the comparison between then can be insert in the discussions session.*

Percentages of susceptibility classes in four landslide prediction maps are already summarized in Table 1. The comparison between them are inserted in to the Discussion section (page 10 lines 24-32) as follows;

*RC1-11   Consequently, to all the required modifications and suggestions the conclusion has to be reviewed accordingly.*

The Conclusion has been rearranged according to the revisions made in the previous sections (page 11 line21 - page 12 line 23)

**Minor issues**

*RC1-12    Add some more recent references about the "landslide-specific information for emergency and disaster management activities in the world". See for example Solari et al., 2018*

Four recent references have been added in to the reference list and cited in the text (page 1 line 23, page 2 line 8, page 7 line 31).

*RC1-13   Add references of already published methods, e.g. IDW, NDVI and LST*

For NDVI and LST, references are already there in the manuscript (page 7 lines 13-14 and lines 22).

IDW is a standard interpolation method and is used to interpolate the rainfall data through out the study area for better computations.

*RC1-14   Pay attention to the tenses. You write some parts using the present form and other the past.*

Tenses have been corrected in the manuscript. (page 2 line 27)

*RC1-15   Please check – Line 24 page 2 - remove "could"*

Deleted. (page 2 line 24)

*RC1-16   Substitute "from the Mean Sea Level" with a.s.l. (above sea level) – Line 26 page 4*

Corrected. (page 4 line 17)

*RC1-17   Line 3 page 5 remove "for these data"*

Corrected. (page 5 line 11)

*RC1-18   Line 7 page 5 - Substitute "an inventory map of landslide" with "landslide inventory map"*

Corrected. (page 5 lines 15-16)

**Response to Anonymous Referee #2.**

*The paper deals with a topic of interest for the journal. I think it could be of interest for the readers. However, in my opinion there is still work to be done in order to make it suitable for publication.*

*RC2-1   As the other reviewers, I think itis not well organized.*

Please see responses to RC1-1,RC1-3, RC1-4.

*RC2-2   The abstract is "strange". It is not a good summary of the paper.*

Please see response to *RC1-2* of the Anonymous Referee # 1,

*RC2-3   I think also that the introduction is not well focused and too long.*

Please see response to *RC1-3* of the Anonymous Referee # 1,

*RC2-4   And I see too long sentences which sometimes makes difficult the understanding. Can you improve it?*

Has been addressed (page 1 lines 23-29) as;

*RC2-5   The quality of the figures is poor. Why do not use colour figures?*

Please see response to *RC1-1* of the Anonymous Referee # 1,

*RC2-6   The analysis of the results is also very qualitative.*

All the predisposing factors are overlaid with the training sample from the landslide failure map and the weight of susceptibility index for landslide occurrences have been calculated. Then by using bivariate and multivariate analysis landslide prediction models are generated with and without radar derived factors. Hence, all the landslide prediction analysis is quantitative. However, in order to make the models are more interpretable for the users, weight of indices is discretised in to four classes as 60%, 30%, 10%, and 0% of failure regions for high, moderate, low, and very low landslide susceptibility classes respectively.

*RC2-7     In the conclusion, the authors say that "with the integration of RIF as surface roughness, near surface soil moisture 15 from Delta Index, and forest biomass, the detection of the boundary between the high and very low susceptibility areas is increased". However, it is not well demonstrated from the given results and explanation. Can you improve it?*

Table 1, Landslide susceptible area comparison from bivariate and multivariate analysis without and with RIF, BiNR -Bivariate analysis without RIF, BiWR -Bivariate analysis with RIF, MNR -Multivariate analysis without RIF, MWR -Multivariate analysis with RIF, describes the particular results and under the Discussions section, the results are explained further (page 13 lines 3-11) Please note that the RIF has been replaced with RDF, for Radar Derived Factors.

*RC2-8 I am not sure that from the result one can conclude that RIF helps to improve the results. I see very similar results by using and by not using the RIF parameters. Please, can you improve your analysis in order to be more convenient or change the conclusion?*

All prediction and validation analysis are based on the past landslide experiences in the same area, thereby minimizing bias and errors from human intervention. In multivariate analysis, weights are calculated by using expert knowledge. However, consistency ratio is measured in order to confirm the consistency of relative importance. Hence, all prediction results are depending on the past landslide occurred in this study area and the statistical analysis. Table 1 compares the landslide susceptible areas from four different landslide prediction models by bivariate and multivariate with and without radar induced (derived) factors numerically. Even though they appear to be similar, I confirmed that all the analyses are numerical, and give different meanings, especially within the context of the study.

**Response to Short Comment #1.**

*The manuscript shows the comparison among different approaches (bivariate/multivariate analyses) using different sets of data (classic/classic + radar data) to produce a landslide susceptibility map of an area located in Sri Lanka. The work in general seems appropriate for the journal but it is not very well organized. In the paper a reader would expect to read: 1) a comprehensive introduction with proper literature,2) a detailed description of the study area and its problematic in terms natural hazard;3) a description of the adopted methodology; 4) the presentation of the results, 5) a discussion of the obtained results; 6) final remarks. I think the manuscript contains some of these issues but not well organized.*

*SC1-1 The introduction session is very long with respect to the rest of the paper.*

Please see response to *RC1-3* of the Anonymous Referee # 1

*SC1-2 The authors should add some background knowledge about the use of remote sensing data and in particular of radar data to infer topographical, soil and land cover information.*

"Use of radar remote sensing for topographical information" is added to page 6 lines 19-23, for soil information in page 8 lines 3-4, and for land cover information in page 8 line 30 -page 9 line1.

*SC1-3   The literature review part in the first part of the Introduction needs to be improved. The second part (Statistical methods for landslide susceptibility analysis) should be reduced and part of it should be moved in to the methodology description.*

The introduction section was improved by reducing the extra information where unnecessary. The following lines , page 2 lines 3-5, lines 10-12, line 22, and lines 28-29 have been deleted.

Further, as commented, a section from the statistical methods (give references) for landslide susceptibility analysis has been moved to the methodology section (give references to the new locations).

*SC1-4   The description of the study area is very short. Please add some information about the geology of the study area and about the typology of the landslides which affect the study area.*

Geological information about the study area has been inserted to page 4 line 31 -page 5 line 4 as;

Typology of the landslides of the particular area has been inserted to page 4 lines 22 – 24 as;

*SC1-5   The section "Data and methodology" is actually a list of the data available. There is nothing about the bivariate or multivariate methods. I suggest to show a map for each considered predisposing factor.*

The Data and Methodology section has been separated, and the methodology section has been improved by making additions to page 9 line 20 – page 11 line 3as;

Statistical analyses of bivariate and multivariate methods have been inserted to page 9 line 22 – page 11 line 3 as;

When considering the guidelines for manuscript preparation, even though the individual figures from fifteen predisposing factors are really significant, it is difficult to add them all to the manuscript. Hence, all the fifteen predisposing factors (in colour figures) were added as supplementary materials (Sup 1-3).

*SC1-6   Some factors need for a more accurate description, for example you need to describe the geology of the study area (Geological factors), in this paragraph information about the geology of the study area and the used classes totally lack.*

The geological information of the study area has been inserted in to the manuscript (page 4 line 31 – page 5 line 4) as;

Additional geological information and used classes have been included under the Geological factors on page 7 lines 27 – 29 as;

"Primarily the undifferentiated charnockitic biotite gneisses and Quartzites are prominent with Garnet-sillimanite and Garnetiferous quartzofeldspathic gneiss in the study area."

*SC1-7   How do you decide the weight of influence of all predisposing factors?*

The details of the relative weight calculation in bivariate, information value method (page 8 lines 8 – 16) and multivariate, MCDA based on AHP is inserted in to the manuscript (page 8 lines 18 – 28).

*SC1-8   I suggest to split the results from the discussion. In the results section you need to present the landslide susceptibility maps and to explain their significance in terms of predisposing factors. In the discussion you can compare all the obtained maps highlighting advantages, drawbacks and limitation.*

The Discussion and the Results have been separated into two, and described accordingly (page 10 line 23).

Under the Results section, four (04) colour landslide susceptibility maps have been inserted to the manuscript by preserving the colour blindness. (Figure 4)

*SC1-9   Figure 1: I think that a colour figure can have more appeal, the same for figure 3.*

Please see response *RC1*-1 to the Anonymous Referee # 1,

**Minor issues**

*SC1-10   Page 1 Line 23: I think that you mean 90% and not 09%*

According to the literature Chalkiaset al., 2014, landslides from all natural hazards are 9% not 90%.

*SC1-11   Page 2 Line 11: Earth and not earth*

When reducing the Introduction section by removing unnecessary information, the particular line has been deleted.

*SC1-12   Page 2 Line 33: delete "employed"*

Corrected. (page 2 line 24)

*SC1-13   Page 4 Line 24: "act as a sponge" does not sound really scientific*

Inserted the word "act as a highly absorbing entity" instead of *"act as a sponge"* (page 4 lines 4-5)

*SC1-14   Page 5 line 5: how much is the DEM resolution?*

DEM resolution has been added in to the manuscript (page 4 line 21).

*SC1-15   Page 7 line 6: what does "Thermal-NDVI space" mean?*

There is a unique relationship between soil moisture, NDVI, and Land Surface Temperature for a given region. This relationship is described as the "Universal Triangle" and results have been confirmed through theoretical studies using soil-vegetation-atmosphere-transfer (SVAT) model (Wang and Qu, 2009, Zenga et al., 2004).

*SC1-16    Page 9 Line 2: How do you extracted the lineaments from Landsat and Sentinel 2 images? Are you sure that joints and fractures can be observed with the resolution of Landsat and Sentinel?*

This study only used 10m resolution Sentinel-2A image (not Landsat) for the lineament extraction of the study area. Most recent studies such as Kati et al., 2018 and Adiri et al., 2017 confirmed the use of 10m resolution Sentinel 1 and 2A images for lineament extractions.

*SC1-17    Several references are not reported in the reference list: (van Vesten 1997; Somaratne, 2016; Rahman et al., 2008; Septiadi and Nasution 2009; Zhan et al., 2002)*

Missing references have been added to the reference list.

[revised manuscript text omitted]

---

## Author Response (AR2)

Dear Editor in Chief

We are pleased to submit a revised manuscript entitled *Efficacy of using Radar Derived Factors in Landslide Susceptibility Analysis: case study of Koslanda, Sri Lanka* for publication in the Journal of Natural Hazards and Earth System Sciences. A revised copy of the manuscript is provided with changes to the manuscript requested by the Referees reports indicated in the attached document, together with detailed responses to the editor's and referees' comments.

Yours Sincerely

AKRN Ranasinghe

**Responses to Anonymous referees reports on the paper " Efficacy of using Radar Derived Factors in Landslide Susceptibility Analysis: case study of Koslanda, Sri Lanka"**

We wish to thank the editor and all reviewers for their constructive comments to arrange a successful journal paper with minor revisions. In the following, the comments of the referees are shown in italics and our responses indented in normal text. References to the edited lines are according to those found in the revised manuscript, unless specifically referred to in the original manuscript.

**Response to Anonymous Referee #2 – Report 1**

Accepted as is

**Response to Anonymous Referee #3 – Report 2**

*The paper requires minor revisions, listed below.*

1) *line 21 pg 1, 9% instead of 09%. Line 24 "16500 deaths and affect 4.5 million people worldwide" in which temporal span?*

   Corrected 09% as 9% (page 1 line 21)

   The temporal span is given as 2000 – 2017 (page 1 lines 22-23)

2) *merge sections 1.1 and 1.2*

   Section 1.1 focuses on Methods for Landslide Susceptibility Analysis and section 1.2 focuses on the Landslide Predisposing factors. These are two different aspects. Merging them together might potential confused the reader. Additionally, Referee #2 has accepted as is. Therefore, we did not merge the sections.

3) *line 2 pg 3 "above Mean Sea Level" m.a.s.l is a more common way to write this*

   Corrected (page 3 line 3)

*4) Figure 1 should be improved. I cannot read the text and there is no legend of the DEM*

Figure 1 improved as suggested (See Figure 1, page 14)

*5) I think that the subsection 3.1 is not needed. The title of the section already contains the word "data" Line 26 pg 3 "prediction" is maybe too much, I would rather use "susceptibility"*

Section 3 is removed and Subsection 3.1 changed as section 3 (page 3 lines 24, 25). All the followed headings and subheadings were changed accordingly.

*6) Figure 2 caption remove "failure", "landslide catalogue" instead of "landslide map"*

Corrected (page 15, Figure 2 caption)

*7) Figure 3, caption "work" instead of "methodological", figure: "Database (or catalogue)" instead of "data base", remove "failure"*

Changed the Figure 3 and caption according to the comments (See figure 3, page 17)

*8) section 4.1 and 4.2, remove the acronyms from the title and if possible make them more understandable*

Corrected as commented (page 8 line 23 and page 9 line 1)

*9) enrich the discussion section, it is quite poor with respect to the other sections of the manuscript*

Enriched the discussion section as suggested (page 9 lines 14-21)

[revised manuscript text omitted]